# *Staphylococcus aureus* Internalization in Osteoblast Cells: Mechanisms, Interactions and Biochemical Processes. What Did We Learn from Experimental Models?

**DOI:** 10.3390/pathogens10020239

**Published:** 2021-02-19

**Authors:** Stefano Stracquadanio, Nicolò Musso, Angelita Costantino, Lorenzo Mattia Lazzaro, Stefania Stefani, Dafne Bongiorno

**Affiliations:** 1Medical Molecular Microbiology and Antibiotic Resistance Laboratory (MMARLab), Department of Biomedical and Biotechnological Sciences (BIOMETEC), University of Catania, 95125 Catania, Italy; nmusso@unict.it (N.M.); lazzclml@gmail.com (L.M.L.); stefanis@unict.it (S.S.); dbongio@unict.it (D.B.); 2Department of Drug and Health Sciences, University of Catania, 95125 Catania, Italy; angelita25costantino@gmail.com; 3Interuniversity Consortium for Biotechnology, Area di Ricerca, Padriciano, 34149 Trieste, Italy

**Keywords:** human osteoblast, MG-63, *Staphylococcus aureus*, internalization mechanisms, bone tissue engineering, 3D bone infection model

## Abstract

Bacterial internalization is a strategy that non-intracellular microorganisms use to escape the host immune system and survive inside the human body. Among bacterial species, *Staphylococcus aureus* showed the ability to interact with and infect osteoblasts, causing osteomyelitis as well as bone and joint infection, while also becoming increasingly resistant to antibiotic therapy and a reservoir of bacteria that can make the infection difficult to cure. Despite being a serious issue in orthopedic surgery, little is known about the mechanisms that allow bacteria to enter and survive inside the osteoblasts, due to the lack of consistent experimental models. In this review, we describe the current knowledge about *S. aureus* internalization mechanisms and various aspects of the interaction between bacteria and osteoblasts (e.g., best experimental conditions, bacteria-induced damages and immune system response), focusing on studies performed using the MG-63 osteoblastic cell line, the best traditional (2D) model for the study of this phenomenon to date. At the same time, as it has been widely demonstrated that 2D culture systems are not completely indicative of the dynamic environment in vivo, and more recent 3D models—representative of bone infection—have also been investigated.

## 1. Background

During the first steps of growth, remodeling and recovery of the bone, different cell types co-exist and cooperate to form the extracellular bone matrix (EBM) [1,2]. Among these, osteoblasts are the cells that form the bone and, together with osteoclasts, preserve its physiological homeostasis [3]. 

Pathological conditions, such as bacterial infections, are responsible for altered osteoblast activity. In detail, surgical procedures, especially in the presence of medical (orthopedic) devices, are responsible for an increased susceptibility of osteoblasts to osteomyelitis [4,5,6,7] and, in this context, *Staphylococcus aureus* represents a frequent intra- and extracellular pathogen [8].

The host–pathogen interaction between osteoblasts and *S. aureus* occurs through the recognition of pathogen-associated molecular patterns (PAMPs) by the pattern recognition receptors (PRRs) exposed on the extracellular surface of the osteoblasts. The consequent production of chemokines and cytokines is responsible for the recruitment and subsequent activation of innate and adaptive immune cells, typical of the cellular inflammatory response [9]. At the same time, the overstimulation of osteoblasts by *S. aureus* causes an increase in osteoclastogenesis with consequent osteoblast death, as well as an alteration of bone homeostasis (Figure 1) [10,11].

The presence of proteins and glycans—such as type I collagen, bone sialoprotein, osteopontin and fibronectin—make the EBM a perfect niche for *S. aureus* that binds these EBM components through multiple adhesins known as microbial surface components which recognize adhesive matrix molecules (MSCRAMMs) [8,12]. Indeed, the *S. aureus* attachment to the EBM represents a key step in the onset of osteomyelitis, where type I collagen represents approximately 90–95% of the organic fraction of the EBM directly interacting with this pathogen (Figure 1).

Recently, it was demonstrated that the ability of *S. aureus* to internalize inside osteoblasts is a key strategy to protect itself and maintain the infection. On the contrary, osteoblasts respond to *S. aureus* internalization by secreting inflammatory factors—such as cytokines, chemokines and growth factors—which, in turn, activate and recruit immune cells from the innate or adaptive immune systems (Figure 1) [13]. 

The excessive formation and activation of osteoclasts is caused by an increase in RANK-L expression and production through the COX-2/PGE2 pathway. This leads to severe bone resorption as well as to decreased production of osteoprotegerin (OPG). Finally, through the release of membrane-damaging virulence factors such as phenol-soluble modulins (PSMs), *S. aureus* can cause osteoblast necrosis and apoptosis through intrinsic and extrinsic caspase pathways. These processes can lead to the release of intracellular *S. aureus*, which can re-infect other osteoblasts. 

Consequently, the ability of *S. aureus* to survive in osteoblasts after internalization also results in effective escape from the antibiotic therapy, which cannot penetrate inside the cells [14,15]. 

The hypothesis that increasing our understanding of the immune response, as well as intensifying the host’s defenses, could be a valuable avenue for developing new anti-infectious strategies dates back several years [16].

The use of murine or human in vitro culture models—including primary cells, induced osteoblasts from pluripotent stem cells and immortalized and malignant cell lines—has allowed a better understanding of osteoblast cell biology during infection processes [17]. 

To date, the progress of research in the field of orthopedic engineering, as well as the development of new therapies and biomaterials, increases the importance of these in vitro models. At the same time, a deeper knowledge of their phenotypic and genotypic status and their differences in relation to primary human osteoblast cells is needed, especially in order to choose the most appropriate experimental model.

In this regard, the results obtained from in vitro infections of osteoblasts grown as two-dimensional (2D) monolayers provided important information on the molecular mechanisms underlying bacterium–host cell interactions. Despite this, these models do not reproduce the dynamic aspects of this interaction, such as the organization of osteoblasts in healthy bone to provide strength and resistance and therefore to respond better to bacterial infections [18,19]. To overcome these limitations, animal models of osteomyelitis were considered the gold standard for the study of bone infections, but the different responses to bacteria between the mouse model and other animal models made the use of these models not exhaustive [20,21,22,23]. From here, more relevant in vitro models that physiologically mimic the human bone microenvironment have been developed and will be discussed in the last section of this review.

### S. aureus vs. Mycobacterium tuberculosis

Before delving into different aspects of *S. aureus* internalization, a comparison between the behavior of *S. aureus* and that of an obligate intracellular bacterium, such as *Mycobacterium tuberculosis*—the etiological agent of tuberculosis (TBC)—could be useful.

While *M. tuberculosis* needs to replicate within human cells to disseminate to other individuals and cause disease, internalization of *S. aureus* by osteoblasts is a key element in the spread of the infection, as it allows *S. aureus* to persist inside osteoblasts protected from the immune system and it gives *S. aureus* the opportunity to sustain the infection [3], but it is not necessary for its replication. 

*M. tuberculosis* spreads from person to person almost exclusively by aerosolized particles that can be trapped in the upper airway or oropharynx. Once in the lower respiratory tract, *M. tuberculosis* is primarily phagocytosed by macrophages and dendritic cells, but neutrophils are also infected [24]. Although *M. tuberculosis* usually infects macrophages, it was also found in non-myelocytic cells of TBC patient. As *M. tuberculosis* internalization in non-phagocytic cells is an actin-dependent process involving heparin-binding hemagglutinin, toll-like receptors (TLRs), surfactant proteins and complement and scavenger receptors [25], *S. aureus* internalization involves some cytoskeletal elements too, particularly actin microfilaments. It has been demonstrated that the internalization process of *S. aureus* can occur with dead bacteria but not with dead osteoblasts, suggesting that the internalization process is more of an active cellular mechanism than an active bacterial mechanism [26].

After *M. tuberculosis* or *S. aureus* internalization, infected cells trigger a local inflammatory response that attracts immune cells to the site of infection. Osteoblasts infected by *S. aureus* secrete inflammatory factors like cytokines, chemokines and growth factors, all of which can activate and recruit immune cells from the innate or adaptive immune systems [27], while *M. tuberculosis* promotes the buildup of cellular aggregates forming the granulomas, that represent a complex environment constituted by macrophages, multinucleated giant cells, epithelioid and foamy cells, granulocytes and lymphocytes [25].

We only reported the main common characteristics regarding the internalization process of both *M. tuberculosis* and *S. aureus*, as *M. tuberculosis* internalization in phagocytic cells needs more tricks to allow the bacteria to escape the phagosomal threat. 

## 2. MG-63 Osteoblast-Like Cell Line as an Effective In Vitro Model to Investigate Host–Pathogen Mechanisms during *S. aureus* Infection

Although preclinical models are known to offer an essential prescreening method for testing new biomaterials useful in the treatment of orthopedic disorders, the increasingly restrictive regulations for the use of in vivo models and the ever-increasing demand for primary cells from healthy or sick donors have led to the development of “continuous” osteoblast cell models. Among these, human- and animal-derived primary cells [28,29,30,31,32], immortalized cell lines [33,34], malignant cells [35,36,37], and induced pluripotent stem cells (iPSs) are used not only in drug and biomaterial testing, but also in bone biology investigations.

Certainly, primary cells, deriving directly from patients, better reproduce the behavior of the original cellular niche, resulting in a preclinical model closer to clinical conditions. Over the years, however, researchers and physicians have realized that the phenotypic and often genotypic differences of these cells isolated from different patients make it difficult to study the disease under examination and consequently to establish the best therapeutic strategy.

An improvement in the knowledge of bone biology and, in particular, osteoblast cells has been achieved through the development of stabilized osteoblast cell lines as models for in vitro investigation of cell differentiation, cytokines and hormone regulation, matrix protein synthesis and secretion and molecular mechanisms of bone diseases. At the same time, these models were found to be useful for the evaluation of the cytocompatibility and osteogenicity of new biomaterials [17].

There is no evidence indicating the superiority of one model over the others; therefore, an evaluation of their respective advantages and disadvantages, on the basis of studies to be conducted, is important.

Primary cultures represent an in vitro model that uses cells directly obtained from tissue biopsies (~1 cm^3^) or organ dissections. These cells have the unique characteristic of maintaining their genetic, morphological and functional features. This makes them the best representative indicators of normal cell phenotype and early-stage disease progression, and as such they are commonly used as in vitro tools for preclinical and investigative biological research and toxicological studies, besides reducing the number of animals required for preclinical toxicology studies at an initial stage, making them cost-saving.

Even though primary human osteoblasts tend to preserve their differentiated phenotype in vitro, after a certain number of cell divisions, these cells have a limited lifespan and will stop dividing (or senesce) and may be more difficult to grow and maintain than a continuous (immortalized) cell line. Induced variability in primary cells obtained from donors and in subculture practices is one of the main challenges faced by researchers studying cell signaling pathways [17]. 

Specifically, it has been shown that the age of the donor influences the proliferative capacity of isolated cells, whose proliferation times are doubled if derived from patients over the age of 65 [38,39].

Furthermore, bone aging, defined as a change in the degree and distribution of bone mineralization, is also age-dependent. This is reflected in the physiology of isolated osteoblast cells, characterized not only by slower proliferation, but also by phenotypic modification [40,41]. Likewise, the expression of genes and the synthesis of proteins associated with the osteoblast phenotype are also influenced by the age of the donor, as well as by the anatomical site of isolation. For example, it is now known that FGFβ and IGFII gene expression is downregulated in osteoblast cells isolated from the mandible, and the synthesis of type I collagen and osteonectin shows higher expression in cells isolated from fetal to 20-year-old bone donors, while a 65% decrease in collagen levels was observed in cells from donors older than 20 years [42].

Moreover, when these cells are extracted ex vivo and transferred to a culture environment, they may lose their structural and functional characteristics. In this regard, cells having completely different morphology in vivo at the tissue level may show similar morphology in the culture environment [43,44].

Thus, donor age, site of isolation and the gender differences that we have discussed so far are just some of the factors that can influence the behavior of primary human osteoblast cells and, in turn, confer different times of phenotypic modification in vitro. As a result, in the absence of a homogenous target of patients/donors, obtaining cultures of osteoblasts suitable for the study of basic applied biology or particular mechanisms, such as infection, is not efficient.

The ease of obtaining results in experiments and their repeatability, as well as the ease of maintenance, the unlimited number of cells without the need for isolation and the relative phenotypic stability of immortalized or continuous MG-63 cell lines has allowed researchers, in some respects, to overcome the limits imposed by cells primarily derived from the bone (HObs). Although these cell models differ in some respects from primary osteoblast cells, Czekanska et al. showed that MG-63 cells show some distinct similarities to HObs [45].

According to Billiau et al., the MG-63 cell line is derived from a juxtacortical osteosarcoma diagnosed in the distal diaphysis of the left femur of a 14-year-old male [35]. When cultured, these cells appear as rapidly proliferating aggregates without exhibiting contact inhibition [46].

The similarity between MG-63 and HObs was already studied several years ago, when Franceschi et al. observed the response of these cells to 1,25-dihydroxyvitamin D3 (1,25 (OH) 2D3) as an effect on cell morphology and on the phenotype comparable to normal osteoblasts [47].

More recent studies have shown that the cell growth kinetics of MG-63 were comparable to that of HObs as the exponential growth phase was observed from day 2 to day 6, followed by a plateau phase from day 6 to day 10 [45]. This result is confirmed by the ability of infinite proliferation typical of malignant cells, such as MG-63 cells, where the lack of intrinsic cell cycle control contributes to tumor progression.

Czekanska et al. also observed that the activity of alkaline phosphatase, an enzyme identifying mature osteoblasts, was lower in MG-63 cells than in primary cells [45]. This result confirmed the different degree of differentiation of MG-63 towards a more immature phenotype, compared to HObs.

The expression analysis of key osteoblast-specific genes [45] showed that the transcription factor Runx2—which regulates gene expression of the all-important bone matrix proteins (including ALP, OC, BSP and type I collagen)—is expressed more in MG-63 than in HObs, except on day 2 [48,49].

As previously introduced, type I collagen is essential for the function of osteoblast cells [50] and is overexpressed in the phase preceding matrix mineralization [51,52], in order to allow the formation of fibrils and a subsequent physiological maturation of the matrix. On the contrary, MG-63 cells show a low expression of type I collagen as well as of osteocalcin [45]. Consequently, the reported studies highlight the limitations of these cells as a model for the phenotypic development of osteoblasts as well as for the evaluation of the mineralization of the matrix and the properties of new biomaterials [45].

On the other hand, the MG-63 cell line proved to be a valid in vitro model for the study of bacterial infection mechanisms, especially during *S. aureus* internalization [14,53,54,55,56,57]. In 2010, Schroder and Tschopp demonstrated that the innate immune response against pathogens involves the activation of an inflammatory pathway known as the inflammasome activation pathway [58]. Inflammasomes are multiprotein signaling complexes that are assembled following the recognition of stress/pathogenic signals; among these, caspase-1 is the most involved [59]. Upon stimulation by pathogens, caspase-1 binds to an adapter molecule known as apoptosis- associated speck-like protein (ASC) [60]. This binding leads to the autocatalytic cleavage of caspase-1, the processing of pro-IL-1β and pro-IL-18 and the secretion of mature IL-1β and IL-18, triggering in some cases even an inflammatory form of cell death (pyroptosis) [61]). Recent studies have shown that this also applies to *S. aureus* and MG-63 cells [62,63]. 

Finally, in previously published works, we demonstrated that internalization, using MG-63 cells, is a pathophysiological pathway of some methicillin-resistant *S. aureus* (MRSA) which depends on the total number of cells infected and not on the number of bacterial cells that enter each osteoblast. Furthermore, even if our strains were not homogeneous in terms of genetic backgrounds and virulence factors, ST22-IVh and ST239-III *S. aureus* showed higher intracellular persistence in host cells, making them more prone to developing chronic and recurrent infections, and the different genetic background was also accompanied by a different modulation of inflammatory phenomena, metabolism and antioxidant machinery [64,65].

**Take home message:** We can conclude that although primary cell lines, and in particular HObs, have the advantage of maintaining their genetic, morphological and physiological features, they also have a limited life span and are difficult to grow and maintain in continuous cultures. MG-63, in our experience, is also a valid in vitro model for the study of *S. aureus* internalization and persistence, both as regards the mechanisms underlying the ability of *S. aureus* to adhere, invade and persist within osteoblasts and the host cell response to infection.

## 3. Multiplicity of Infection (MOI) and Invasiveness of Different Bacterial Strains

In the presence of prosthetic devices, complete eradication of bacterial infection is often a challenging task. Internalization in non-professional phagocytes is an important pathogenic mechanism actuated by bacteria to elude host defenses and medical therapies. The efficiency of invasion differs across bacterial species and adjustments to the titer of the microbial *inocula* used in the assay are often needed to enumerate intracellular bacteria. 

There is a precise relation between the inoculum, in terms of multiplicity of infection (MOI), and the internalized bacteria. Furthermore, there is a relationship between MOI, internalized bacterium ratio and medical therapies [66,67].

Intracellular invasion occurs through a variety of pathogenic species. Some bacteria are obligate intracellular pathogens, while other only become intracellular to escape the host immune system. Among these, the following genera are the most representative: *Mycobacterium; Escherichia; Salmonella; Listeria; Shigella; Legionella; Chlamydia; Yersinia; Streptococcus; Staphylococcus* and *Enterococcus* (in particular *E. faecalis*). *S. aureus* is the only one capable of causing the onset of clinically relevant pathogenic mechanisms and consistently invades osteoblast cells [68].

Different MOIs have been adopted depending on the bacterial species tested. For example, an MOI of 100:1 (bacteria:host cells) is the inoculum titer most frequently used to test *S. aureus*. The inoculum level increases to 500:1 for *S. epidermidis* [55,69] and 1000:1 for *S. lugdunensis* [70]. The different MOIs used showed rapid and efficient internalization of *S. aureus* at low inoculum levels and inefficient internalization of other species at high inoculum levels.

Two other parameters are used to express the potential of invasiveness of bacterial strains: (i) the number of internalized bacteria (NIB) at an established MOI, expressed in term of Colony-Forming Units (CFUs) per number of eukaryotic cells, influenced by the MOI used; and (ii) the percentage of internalized bacteria (PIB), which represents the percent fraction of the inoculum taken up by eukaryotic cells; this value is not affected by the MOI and can be used to express the degree of invasiveness of prokaryotic cells into eukaryotic cells.

Examining the correlation between MOI values and PIB values, it emerged that—over a broad range of inoculum levels—the MOI did not appear to affect PIB values [71]. PIB values can be used to compare strain invasiveness without fearing major effects resulting from varying MOIs.

However, a new parameter was proposed to express the invasiveness of bacterial strains: the internalization minimal inoculum (IMI), corresponding to the lowest MOI required for the internalization of a single bacterium. This value is inversely related to invasiveness and corresponds to the lowest concentration at which internalization occurs under the test condition used. Internalization at a 1:1 MOI inoculum (l1M) corresponds to the number of bacteria internalized when hypothetically exposing each eukaryotic cell to a single bacterium (i.e., using a 1:1 MOI). Its value is proportional to the degree of invasiveness of the strain given by the log10 of the IMI value (LIMI), obtained from the regression curve of log MOI vs. log (CFU). 

In conclusion, the most used parameters to express the intracellular invasiveness of bacterial strains are the NIB and PIB values. PIB values can be used across a broad range of MOIs without fearing the influence of the inoculum size. Ultimately, PIB values do not depend on the MOI, whereas NIB values are strongly MOI-dependent.

Therefore, we can speculate that the internalization process can be influenced by several factors, such as: (i) bacterial sedimentation rate, influenced by the microbe size, shape and tendency to agglomerate and by the viscosity of the medium; (ii) composition of the culture medium; (iii) cell line used in the assay, considering its histological origin, phagocytic activity (professional or non-professional phagocytic cells in primary or secondary cultures), level of expression of integrins capable of interacting with adhesins; (iv) bacterial strain type (bacterial species and genes encoding for invasiveness).

As already mentioned, different MOIs have been used in several studies depending on the strains tested. These include e.g., *S. aureus*, *S. epidermidis* and *S. lugdunensis*, opportunistic pathogens causing implant-related infections. 

These species can survive antibiotic therapies through different mechanisms related to genetic determinants, biofilm production and penetration into eukaryotic cells as the main causes of chronic infections. Furthermore, eukaryotic cells are impermeable to many antibiotics, such as rifampin, that are able to pass through the prokaryotic cell membrane [72,73].

Many staphylococcal species, other than *S. aureus*, are emerging as opportunistic pathogens capable of causing serious and persistent implant-associated infections. According to some authors, *S. epidermidis* is the foremost isolated staphylococcal species responsible for orthopedic infections and is able to internalize into osteoblasts, whilst others report *S. epidermidis* as the second staphylococcal species isolated during orthopedic infections (Khalil et al. 2007; Valour et al. 2013), but its antimicrobial resistance profile is usually not as severe as that of *S. aureus*. *S. lugdunensis* is an emergent pathogen responsible for periprosthetic infections [74].

The invasive potential of different bacterial species and their ability to internalize into the MG-63 cell line was evaluated using a method based on microtiter plates, where they were challenged with osteoblasts.

Campocia et al., in 2016, used different MOIs for each *Staphylococcus* species tested. The MOI value was always recorded in order to know the inoculum levels actually reached: MOI 560:1 for *S. epidermidis* and MOI 1844:1 for *S. lugdunensis*.

*S. epidermidis* has an extremely low rate of internalization, not comparable with that observed for *S. aureus*. Furthermore, the bacterial survival rate appeared rather marginal. Most *S. epidermidis* tested with MOI 500:1 showed a relatively low internalization (<50 CFUs), while other strains showed high internalization (>50 CFUs). 

Some groups of bacteria appear homogeneous in terms of CFUs internalized regardless of the inoculum level, while others exhibit some heterogeneity in spite of similar inoculum levels. *S. lugdunensis* showed very low levels of internalization regardless of the level of inoculum, even though it was tested with a relatively high MOI (1000:1). 

The species considered exhibited marginal rates of internalization compared to *S. aureus*, since *S. aureus* showed a higher rate of internalization at a lower MOI (100:1).

*S. aureus* requires a very low inoculum to reach a high internalization rate, whereas *S. epidermidis* cell invasiveness remains low and marginal. This finding suggests that the active mechanisms of invasion exhibited by *S. aureus* are either absent or much less efficient in *S. epidermidis*. Similarly, the clinical isolates of *S. lugdunensis* showed a low level of internalization (Figure 2) [70].

To confirm the use of diverse MOIs in the internalization process, studies performed by Valour et al. (2013) compared the different behavior of *S. aureus* and *S. epidermidis* during internalization. As previously shown, *S. epidermidis* has a low rate of internalization, being an innocuous commensal of the human skin and mucous membranes, but it is considered a leading opportunistic pathogen [70].

The contrast between the low incidence of *S. epidermidis* orthopedic device infection and the highly prevalent *S. epidermidis* carriage suggests that *S. epidermidis* bone and joint infections may either correspond to accidental events due to colonizing strains or to a specific, more virulent sub-population of commensal isolates.

Two predominant mechanisms have been proposed to be involved in orthopedic device infections, i.e., bacterial invasion and persistence in non-professional phagocytes, such as osteoblasts [75]; and the bacterial ability to form biofilm [76,77].

To verify the ability of internalization of *S. epidermidis* into the MG-63 cell line, an invasion assay of *S. epidermidis* was carried out using an MOI of 500:1 for *S. epidermidis* and an MOI of 100:1 for *S. aureus* (used as a control strain). 

The results of this assay demonstrated that the number of internalized bacteria was MOI-dependent. There was a difference in bone cell invasion rates between *S. epidermidis* and *S. aureus* strains. *S. epidermidis* showed a lower rate of internalization. This could be due to several factors, such as the “cell line effect”—i.e., the use of the MG-63 cell line—and the acquisition of some phenotypic characteristics that may not reflect the in vivo reality [78]. 

To exclude a bias due to a “cell line effect”, the low internalization rate of *S. epidermidis* was confirmed using primary bone cells. For this reason, all assays were repeated using primary human osteoblasts [69].

Fibronectin-binding protein-like molecules are absent in *S. epidermidis*, therefore, the process of invasion is different from that of *S. aureus*.

Finally, the internalization of *S. epidermidis* in human osteoblasts is not a common pathophysiological mechanism in orthopedic device infections, contrary to what was observed in other clinical situations or with other strains (e.g., *S. aureus*).

### MOI Values Were Selected Depending on the Strain Used and Several other Factors

In general, the best choice is to use as few bacteria as possible to reduce cell damage, as important strain-dependent differences may be missed if extended incubation periods or large inocula are used [79].

Hamza et al. (2014) performed an infection experiment using rat osteoblasts and *S. aureus* at different MOIs over different incubation times. They found that intracellular CFUs increased from MOI 100 to MOI 500 and that MOIs greater than 500 did not result in an increase in intracellular CFUs. Osteoblast viability did not change significantly in an MOI range of 100–1000. As a result, high intracellular CFUs and high osteoblast viability were reached at MOI 500 [80].

In the study carried out by Bongiorno et al. (2020), the frequency of internalization was evaluated in a cell culture model of infection using *S. aureus* and MG-63 osteoblasts at an MOI of 100:1. In order to assess this MOI, they first tested MG-63 infection with *S. aureus* ATCC 12598 at the following MOIs: 12:1, 50:1, 100:1 and 200:1. It was observed that, at MOI 12 and 50, the ability of *S. aureus* to internalize into non-specialized cells, such as the osteoblasts, was very limited, while with an MOI of 200, MG-63 cultured cells showed phenomena of cytotoxicity [64]. 

**Take home message:** Taken together, this information suggests that researchers should choose the right MOI carefully when designing an internalization experiment, strictly depending on the bacterial species (sometimes even the clone) and the cell line. Higher is not always better. It is critical to know if the bacteria are obligate or opportunistic intracellular species and if the cells are professional phagocytes. Furthermore, internalization experiments should consider other, less used, parameters, i.e., number of internalized bacteria (NIB), percentage of internalized bacteria (PIB) and internalization minimal inoculum (IMI), as these can help researchers to better describe and compare their results. 

## 4. Interaction between *S. aureus* and MG-63 Osteoblast Cells

*S. aureus* is capable of inducing DNA damage in several host cells, such as osteoblast-like MG-63 cells. The pathogens develop multiple strategies to promote infections [81], interfering with survival pathways [82] and suppressing the immune response of the host, thus facilitating the establishment of chronic infections and promoting host cell transformations [83]. 

### 4.1. DNA Damage

Bacteria can damage the host DNA directly and indirectly, e.g., through the production of reactive oxygen species (ROSs). *S. aureus* induces disease especially during chronic infections and the chronification of *S. aureus* infection leads to a phenotypical adaption from a highly virulent to a less virulent form called “small colony variant” (SCV), characterized by increased intracellular persistence, diminished ability of immune system stimulation and lower ability to induce low levels of cytokines release [84]. 

*S. aureus* versatility during infection is due to the production of many factors, the most notable being: (i) pore-forming toxins; (ii) exfoliative toxins, involved in tissue disintegration; (iii) adhesins, involved in tissue colonization; (iv) ROS, that can lead to the formation of deleterious oxidative host DNA lesions (promoting the oxidation of guanidine-forming 7,8-dihydro-8-oxoguanine or 8-oxoG); (v) cyclomodulins, which alter the host cell cycle to promote infections.

Deplanche et al. (2019) demonstrated that *S. aureus* induces ROS-mediated DNA damage, followed by DNA repair, and identified “phenol-soluble modulin” (PSM alpha) and lipoproteins (Lpls) as the effectors of this phenomenon. Consequently, 8-oxoG is more expressed in the infected cells.

H2AX is a protein used as a marker of DNA damage. In particular, Deplance et al. demonstrated that *S. aureus* induces dose-dependent H2AX phosphorylation (γH2AX) in osteoblast-like MG-63 cells, in the presence of double-strand break (DSB) damage without apoptosis.

In response to the bacterial agent activity, host cells promote the formation of highly cytotoxic ROS as a defense mechanism against bacteria. 

The role of ROS in causing DNA damage was investigated by incubating host cells with N-acetyl-cysteine (NAC), a ROS inhibitor. The incubation of host cells with NAC for 1 h, 6 h and 20 h before infection with *S. aureus* prevented the induction of DNA damage, showing that ROS are involved in *S. aureus*-induced DNA damage [85].

8-oxoG DNA lesions are the most common type of lesions that can generate DNA double-strand breaks when occurring during DNA replication and are thus deleterious [86]. Eukaryotic cell DNA damage may reversibly arrest cell cycle progression to allow DNA repair [87,88]. In addition to DNA damage, cell cycle arrest may be associated with the actin state organization [89,90].

*S. aureus* triggers ROS-mediated DNA damage, thus affecting the genomic integrity and/or regulating gene transcriptional activation. The induced DNA damage depends on the balance between the levels of the expression of PSMα and Lpls and on bacterial adaptation during chronification, linked to the maintenance of the host genome integrity.

Furthermore, previous studies have proven that the *S. aureus* virulence factors PSMs and Lpls had properties similar to cyclomodulins since they induce G2/M transition delay in infected cells [91,92]. The consequences of *S. aureus*-induced G2/M delay were investigated: the G2 cellular phase is advantageous for bacterial intracellular replication and is associated with a decreased production of antibacterial peptides that may contribute to the persistence of the infection [91,93].

### 4.2. Virulence Factors

As already known, persistent infections are associated with a wide plethora of virulence factors regulated by the “accessory gene regulator” (*agr* system). 

Valour et al. (2015-b) showed that Methicillin-sensitive *Staphylococcus aureus* (MSSA) internalization rates inside osteoblasts were significantly higher in chronic bone and joint infection (BJI) isolates than in acute BJI isolates, and that no difference existed between the two groups in terms of cytotoxicity. Similarly, no differences in the ability of both groups to convert to the SCV phenotype were observed, and biofilm formation was not different between acute and chronic BJI isolates either.

Delta-toxin-negative strains tend to be more represented in chronic BJI and the absence of delta-toxin expression was associated with higher internalization rates. In the same study, the lack of a relationship between BJI chronicity and bacterial genetic backgrounds or virulence factors, as well as an association between osteoblast invasion and *agr* deficiency, were reported. Moreover, acute infections are usually associated with a functional *agr* system. *agr* dysfunction appears to occur during infection and in the presence of persistent bacteremia. Furthermore, infection chronicity appeared to be the main factor associated with *agr* dysfunction. A strong relation between *agr* dysfunction and the bacterial phenotypic mechanism associated with BJI chronicity—including enhanced biofilm formation and increased osteoblastic invasion with reduced infection-induced cytotoxicity—was revealed. The loss of *agr* function that occurs during certain infections seems to be linked with BJI chronicity through the promotion of an intraosteoblastic *S. aureus* reservoir caused by a limitation of intracellular staphylococcal cell damage and through enhanced biofilm formation [16].

Finally, the *agr* system controls the expression of PSM-encoding genes (PSMα 1 to 4, PSMβ 1 and 2 and δ-toxin, sometimes referred to as PSMγ) [94,95]. PSM stimulates the production of inflammatory cytokines [96] and has a role in IL-1β production by infected MG-63 cells. This was demonstrated by analyzing a Wild Type (WT) strain of *S. aureus* and an *S. aureus* LAC Δpsmαβhld mutant for their ability to stimulate the release of IL-1 β. The level of IL-1 β was lower in the supernatant of WT MG-63 cells exposed to the LAC Δpsmαβhld mutant compared to the level observed in the supernatant of WT MG-63 cells exposed to wild-type LAC (Figure 3).

### 4.3. Immune System

After pathogen invasion, the immune system (IS) recruits an inflammasome, an immune signaling platform that activates proteases, such as caspase-1, that proteolytically matures and promotes the secretion of mature IL-1β and IL-18.

The innate immune response against microbes involves an inflammatory pathway known as the activation of inflammasomes. Caspase-1 is synthesized in cells as a 45 kDa inactive precursor that is cleaved to reach its mature form consisting of two subunits of 20 and 10 kDa after inflammasome activation [97]. After stimulation by pathogens, inflammasome assembly leads to the autocatalytic cleavage of caspase-1, the processing of pro-IL-1β and pro-IL-18 and the secretion of mature IL-1β and IL-18.

It was reported that the expression of inflammasome-associated proteins was significantly higher in infected bones than in uninfected ones, as found in patients with osteomyelitis. In an *S. aureus*-induced murine osteomyelitis model, a higher expression of these proteins was reported [98].

Lima Leite et al. (2020) used the MG-63 cell line as a model of infection with an *S. aureus* strain and a CASP1−/− mutant MG-63 cell line (obtained by using the CRISPR-Cas9 editing system) to evaluate the role of caspase-1 after the invasion process. To test caspase-1 activation, MG-63 cells were incubated with bacterial lipopolysaccharides (LPS) and adenosine triphosphate (ATP), two inflammasome activators. Western blot analysis showed that incubating MG-63 cells with the activators resulted in the activation of caspase-1, while the ELISA test revealed a production of low levels of IL-1 β after 2 h and a higher production after 6 h (Figure 4). 

Inflammasome recruited and activated pro-caspase-1, which promoted IL-1β and IL-18 maturation. Six hours after the beginning of the infection with a 50:1 MOI, the number of *S. aureus* CFUs recovered from mutant cells was significantly higher than those recovered from WT MG-63 cells. WT MG-63 cells and CASP1−/− mutant MG-63 clones expressed apoptosis-associated speck-like protein, while only WT MG-63 produced IL-1β after exposure to inflammasome [62].

To evaluate the role of *S. aureus* as an inflammasome activator, IL-1β production in WT and CASP1−/− mutant MG-63 cells was measured. In the latter case, a lack of IL-1β was recorded (in contrast to what was observed in WT MG-63 cells). During these experiments, several *S. aureus* strains were used. All strains induced IL-1β release, showing that this mechanism is strain-independent. Moreover, IL-1β was not detected in the supernatant of MG-63 cells exposed to killed bacteria, suggesting that factors associated with viable bacteria are involved in inflammasome activation. 

*S. aureus* clearance by osteoblast-like MG-63 cells depends on caspase-1. Indeed, the number of viable bacteria recovered from infected cells was significantly larger in CASP1−/− mutant MG-63 than in WT MG-63. This evidence suggest that the lack of caspase-1 impairs the ability of osteoblast-like cells to limit *S. aureus* growth. A drastic increase in the proliferation of internalized bacteria in osteoblastic mutant cells has already been highlighted [62] (Figure 5).

In fact, a correlation exists between the lack of caspase-1 activation and a failure in limiting *S. aureus* replication inside phagocytic cells [99,100]. Furthermore, it was demonstrated by Dinarello et al. (2012) that human osteoblast-like MG-63 cells induce an immune response against *S. aureus* through inflammasome activation and the processing of IL-1β, the main inflammatory cytokine. Finally, CASP1−/− mutant MG-63 cells’ inability to limit the intracellular replication of *S. aureus* was reported. This work points out that active caspase-1 prevents exacerbated intracellular replication of *S. aureus*. Osteoblasts therefore are not passive bystanders, but active players in host defenses against *S. aureus* infection [63].

As mentioned above, *S. aureus* is capable of inducing the production of cytokines and chemokines by binding to extracellular or intracellular receptors. In this way, *S. aureus* induces inflammatory cell recruitment, leading to bone loss [3]. Cultured osteoblasts infected with bacteria secrete immune modulators of the inflammatory response, cytokines and chemokines, which trigger bone inflammation and destruction [65,101,102]. 

To counteract internalization in osteoblasts and the resulting inflammatory process, serratia-peptidase (SPEP)—a metalloprotease produced by *Serratia marcescens* already used as an anti-inflammatory agent—was used [103]. This molecule modulates adhesin expression, enhances antibiotic efficacy toward biofilm-forming bacteria and interferes with *S. aureus* adhesion to abiotic surfaces [104,105].

Selan et al. (2017) showed the effect of SPEP during the internalization process of *S. aureus* in the MG-63 cell-line, using an MOI of 30:1.

The internalization efficiencies of SPEP-pretreated bacteria and untreated bacteria were compared. SPEP-pretreated *S. aureus* exhibited significantly reduced efficiency of internalization. MG-63 cells incubated in the absence of bacteria and with/without SPEP treatment showed that MG-63 proliferation remains unaffected, while when bacteria were pretreated with SPEP, a slight, statistically non-significant decrease in proliferation was recorded for all *S. aureus*. The authors highlighted that the production of chemokines was significantly diminished following treatment with the anti-inflammatory molecule. Chemokine levels in the supernatant derived from MG-63 cells infected with *S. aureus* and pretreated with SPEP were slightly lower than in the supernatant derived from MG-63 cells infected with untreated bacteria. This is a consequence of the reduced internalization of SPEP-pretreated bacteria: lower internalization leads to less stimulation and lower production of the pro-inflammatory chemokine MCP-1 [106].

**Take home message:***S. aureus* interaction with MG-63 is widely studied and, in particular, three aspects are of fundamental importance: DNA damage or mutation, the production of virulence factors and the immune system response. *S. aureus* can damage the host DNA by inducing ROS production and can modify its DNA to adapt to the intracellular environmental (SCV). Moreover, persistent infection is modulated by the *agr* locus responsible for the virulence factors and implicated in the establishment of chronic and persistent infections. After pathogen invasion, host cells prompt the IS to produce the inflammasome and activate an immune signaling platform. 

## 5. Antimicrobial Activity against Intraosteoblastic Pathogens

*S. aureus* can invade osteoblastic cells, which evade the immune response of the host and become a reservoir of bacteria, somewhat protected from the activity of many antimicrobial molecules.

Valour et al. (2015-a) evaluated the efficacy of antimicrobial therapy in *S. aureus* BJI. They evaluated the intraosteoblastic activity of the main antimicrobial agents used for staphylococcal BJI in an in vitro model of osteoblast infection. An infection assay with an MOI of 100:1 was employed, incubating all cells with three different concentrations for each antibiotic.

Inside the bones, vancomycin and daptomycin reach concentrations that cannot significantly prevent bacteria intracellular growth, while an intracellular bacteriostatic effect was observed using ceftaroline and teicoplanin. A significant intracellular bactericidal effect was observed for fosfomycin, linezolid, tigecycline, oxacillin, rifampin, ofloxacin and clindamycin. At the minimum concentration, only rifampin, ofloxacin and fosfomycin were bactericidal. At the maximum concentration, all aforementioned antibiotics were bactericidal, with the exception of vancomycin and daptomycin. Furthermore, at an intracellular concentration, the number of SCVs significantly decreased in the osteoblasts treated with ofloxacin, rifampin and daptomycin. In addition, oxacillin, ceftaroline, linezolid, fosfomycine and tigecycline reduced the proportion of intracellular SCVs, but only at their maximum concentration.

Considering that intraosteoblastic *S. aureus* constitutes a bacterial reservoir leading to chronicity and relapse, targeting intracellular bacteria might be a major therapeutic issue in antimicrobial therapy for BJI [16,107].

Vancomycin intracellular efficacy is lower than that of teicoplanin, probably due to its slow uptake and accumulation in the cell. Conversely, rifampin and fluoroquinolones have a fast intracellular uptake. Previously published studies showed that the intracellular activity of antistaphylococcal drugs depends on the exposure time and extracellular concentration of the molecule tested, which emphasizes the importance of using a therapeutic bone concentration. When using systemic therapeutic concentrations, only ofloxacin was able to limit the intracellular emergence of SCVs. 

A combination of levofloxacin and rifampin is the elective treatment for acute staphylococcal BJI managed with debridement, antibiotics and implant retention (DAIR) [108,109]. This regime is bactericidal and highly active against biofilm-embedded staphylococci and has good bioavailability and bone diffusion [110].

Meléndez-Carmona et al. (2019) evaluated the effects of rifampin and levofloxacin, alone and in combination, against the process of MSSA internalization in MG-63 cells using an MOI of 100:1. Both antibiotics showed a significant CFU decrease (a log10 of CFU) compared to bacterial CFUs within untreated cells, whereas the combination did not show higher activity compared to levofloxacin monotherapy. Rifampin, alone and in combination with levofloxacin, showed a significant increase in the percentage of SCVs and a significant reduction in the number of intracellular CFUs in comparison with untreated osteoblast cells [111]. 

Dupieux et al. (2017) demonstrated that, when using *S. aureus* strains to infect osteoblasts, daptomycin did not reduce MSSA number and was poorly active against MRSA. Instead, oxacillin and ceftaroline revealed significant intracellular activity, although oxacillin is not usually active against MRSA. In this paper, two different molecular combinations were used, in particular: daptomycin/oxacillin was more active against intracellular MSSA and MRSA compared with daptomycin and oxacillin alone; and daptomycin/ceftaroline was less efficient than ceftaroline alone. It seems that in acid intracellular conditions, oxacillin was able to enhance daptomycin activity versus *S. aureus* [112].

Abad et al. (2019) demonstrated that linezolid and tedizolid, in intracellular conditions, were able to slightly reduce the inoculum of *S. aureus* and this reduction was strain-dependent, not MIC-dependent (Minimum Inhibitory Concentration dependent), but improved cell viability. These two oxazolidinones alone are not useful versus *S. aureus* strains associated with chronic forms of BJI due to their weak intracellular activity, but they are able to reduce infection-related cytotoxicity, suggesting a role in modulating the intracellular expression of staphylococcal virulence factors [113].

**Take home message:** Overall, the use of antimicrobials to fight bone and joint infections (BJIs) seems to be more an art than a science, due to the different ability of antibiotics to enter osteoblastic cells, resulting in varying therapy efficacy as well as in the possible failure of molecules which are commonly effective against staphylococcal infections (e.g., vancomycin). In many cases, a combination of two molecules is the right choice to eradicate the infection, but even single molecules that cannot enter cells (e.g., oxazolidinones) can provide important effects by inhibiting, to some extent, the consequences of an infection. 

## 6. Biomimetic 3D In Vitro Models to Investigate Osteomyelitis

In the previous sections, we have seen how conventional models, in particular the use of immortalized cultures such as the MG63 cell line, make a valuable contribution to understanding the mechanisms of host–pathogen interactions, especially those concerning the internalization of *S. aureus* in osteoblasts. Using these models, it was possible to recreate some aspects of osteomyelitis, such as the formation of biofilms or the interaction of bacteria with one or more of the host organism’s cell types [114], but they are far from resembling bone tissue and from reproducing fibrous encapsulation- or osteomyelitis-induced bone abscesses with a necrotic core. 

Nevertheless, it has been widely demonstrated that *S. aureus* can reach the bone or metal implant surface by binding to extracellular matrix (ECM) proteins via microbial surface components that recognize adhesive matrix molecules such as collagen-binding protein and bone protein-binding sialoprotein [115]. Among the numerous survival strategies used, *S. aureus* can proliferate and form microcolonies known as staphylococcal abscess communities (SACs) [116], responsible for an abscess structure with surrounding fibrin deposits which make the bacterial core inaccessible to the host’s immune cells [117]. Furthermore, Flemming et al. recently analyzed the different causes of antibiotic resistance in *S. aureus* when residing in biofilms [118].

The in vivo mouse models currently in use seemed to be able to give a greater contribution to these studies [22], but have several flaws, for example, planktonic bacteria are inoculated directly into the bone, without taking into account that the origin of osteomyelitis is often derived from a biofilm that bacteria have formed at the site of infection. In addition, there is an incompatibility between the results obtained in mice and those observed in patients [23].

In this regard, the opportunity of developing a sophisticated 3D in vitro model able to recreate the dynamic changes in osteomyelitis infection is of great interest.

Raic et al. developed a 3D in vitro model of biofilm-induced osteomyelitis to study the effects of postoperative osteomyelitis-inducing bacteria on the bone marrow [119], as the analysis of this tissue allows the study of the early stages of the infection, which is not clinically apparent and therefore difficult to treat [1,120,121]. Their system is the first biomimetic human in vitro osteomyelitis model to allow understanding of the early stages of disease progression and to overcome the limitations of other model systems, for example: (a) results of animal models are often not transferable to human beings [20,122], while their bone marrow model does not pose the problem of interspecies-related differences, as it is composed of human cells; (b) in vitro models are for the most part performed on conventional 2D tissue culture plates that are not able to mimic the natural 3D conditions, potentially generating in vitro artifacts; (c) to mimic biofilm-triggered osteomyelitis [23,123], especially in animal studies, planktonic bacteria are used and this does not reflect the in vivo situation; contrary to that, biofilms of planktonic bacteria are used in this model to mimic biofilm-triggered osteomyelitis. In detail, postoperative osteomyelitis was reproduced by developing a new 3D protein scaffold with a macroporous architecture that resembles the trabecular bone. In order to mimic the cellular compartment of the stem cell niche, human hematopoietic stem and progenitor cells (HSPCs) and mesenchymal stromal cells (MSCs) were seeded within this scaffold.

A very important component for the development of these models is the choice of the material to be used. In 2010, Pagedar et al. showed that the level of biofilm formation of *S. aureus* depends on the hydrophilicity of the surface [124]. In fact, the model developed by Raic in titanium (a material used in the clinic) showed the formation of biofilms containing active metabolic bacteria different from the conventional plastic plates used for in vitro cell cultures [119]. 

As demonstrated by Meng et al. in 2014, followed by studies by Ravi et al., cells exhibit different behaviors in terms of differentiation, protein expression or cell survival rate when grown in a 2D or 3D system [125,126]. Similarly, these cells will react differently to bacterial factors when placed in a 2D or 3D environment. This suggests the importance of carefully selecting not only the materials, but also the bacteria that are relevant for in vitro functional modeling of osteomyelitis. Moreover, the use of these 3D models may provide a better understanding of the molecular interactions and cellular responses to osteomyelitis, which are crucial for the development of new therapies for the treatment of this debilitating disease. In this regard, Kavanagh et al. developed a three-dimensional (3D) bone infection model to examine the processes of *S. aureus* bone colonization and infection [127]. To simulate the infection process, the scaffold, produced using an EDAC (1-Ethyl-3-(3-Dimethylaminopropyl)carbodiimide) cross-linked glycosaminoglycan collagen biomaterial, was inoculated with a co-culture of osteoblasts and *S. aureus*. Thanks to this model, it was possible to observe the ability of osteoblasts to counteract bone loss and bone destruction by increasing the levels of mineralization of the weakened bone. This discovery is groundbreaking and is only observable in a 3D environment. Indeed, in stark contrast to what has just been described, the same authors had previously shown that in 2D cell culture conditions, *S. aureus* protein A mediates attachment to osteoblasts, but following this link, there was a loss of proliferation and the inhibition of mineralization in osteoblasts [9,11,128]. The development of a physiologically more relevant collagen-based scaffold not only has given a new insight into this pathological phenomenon, but is in line with the characteristic signs of osteomyelitis observed clinically. The images of the two models described above are shown in Figure 6. For a more exhaustive discussion of 3D host–pathogen infection models compared to conventional systems, also refer to Hofstee et al. [114] and Barila et al. [129].

**Take home message:** In conclusion, osteomyelitis infection is characterized by a complex and dynamic environment that cannot be fully understood using a conventional model. Thus, physiologically more relevant collagen-based scaffolds represent an innovative and valuable tool to investigate the ability of osteoblasts to counteract bone loss and bone destruction by increasing the levels of mineralization of weakened bone in a similar way to what happens in patients.

## 7. Conclusions 

For over half a century, the study of the mechanisms that govern osteomyelitis has remained the weak point of orthopedic surgery. Despite many advances in understanding the pathophysiological consequences of bone infection, standards of care treatments have not undergone major changes and some aspects of intracellular persistence in chronic osteomyelitis have not yet been well elucidated [130].

As demonstrated in several publications, biofilm-forming *S. aureus* is the most common pathogen in implant-associated infections, as well as the main cause of reinfection, due to its high resistance to the immune response and antibiotic treatments [131,132,133,134]. Our knowledge about the ability of *S. aureus* to infect not only bone-forming cells (osteoblasts) but also the cells responsible for bone resorption (osteoclasts) [135] derives from the use of conventional in vivo and in vitro 2D models. In particular, in this review, we have analyzed how the use of a standardized in vitro model, such as immortalized MG-63 cell cultures, provides valuable help in understanding the mechanisms of internalization and interaction of *S. aureus* and osteoblasts. Despite being different from primary osteoblasts in some respects, these cells allow us to standardize studies aiming to obtain further information to predict the capability of staphylococcal clones—often associated with recurrent and chronic infections—to invade, internalize and persist within the human cells, as well as to confirm the active role of osteoblasts in the host defense against *S. aureus* infections. At the same time, these models have recently been defined as physiologically limited systems and unable to mimic the complex dynamic environment in which cells are found in the human body [18,19,136]. To make up for this, 3D models (scaffolds) consisting of engineered biomaterials capable of reproducing bone tissue have been developed and it was demonstrated that they can provide information otherwise not obtainable through traditional models. It is clear that conventional 2D in vitro models represent a valid model for studying some of the aspects that occur during bacterial infection (for example, the use of MG-63 cells for the internalization mechanisms of *S. aureus*) in detail. Conversely, 3D bone scaffolds ensure a dynamic and global view of the pathological phenomenon of interest (for example, tissue–host–pathogen interactions).

## Figures and Tables

**Figure 1 pathogens-10-00239-f001:**
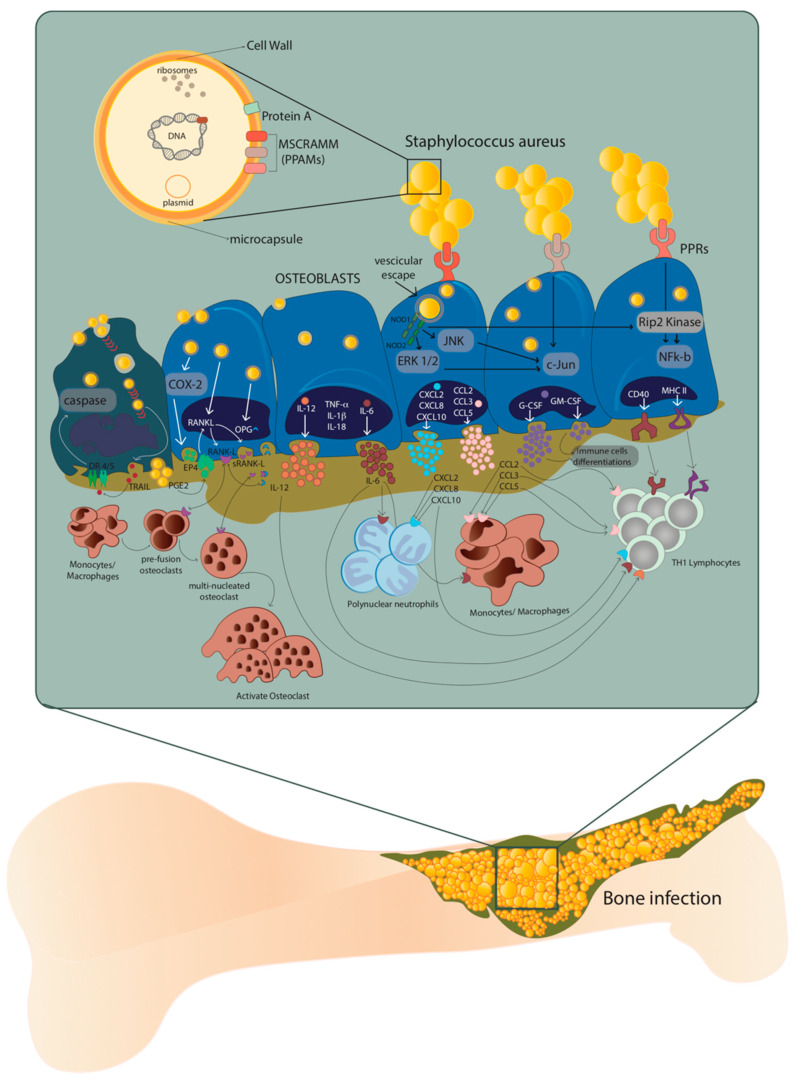
Host–pathogen interaction between osteoblasts and *Staphylococcus aureus*. After internalization, *S. aureus* escapes from the vesicle and interacts with extracellular receptors TLR2 and TNFR-1, as well as with intracellular receptors TLR9 and NODs through α5β1 integrin and actin filaments of the osteoblasts. This interaction increases the expression of cytokines IL-1β, IL-18, TNF-α, the production and release of IL-6, IL-12 and the expression and release of chemokines CXCL2, CXCL8, CXCL10, CCL2, CCL3, CCL5 and growth factors G-CSF and GM-CSF. At the same time, the expression and production of CD40 and MHC II increase. Illustration by A. Costantino (co-author).

**Figure 2 pathogens-10-00239-f002:**
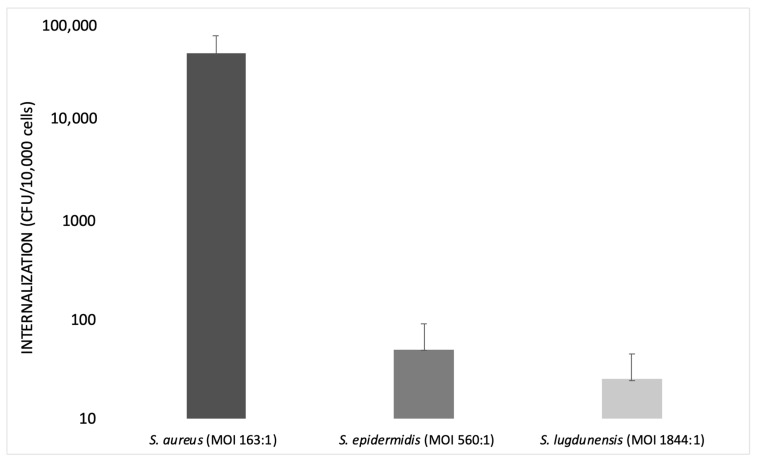
Bar graph illustrating the internalization of *S. aureus*, *S. epidermidis* and *S. lugdunensis* at different multiplicities of infection (MOIs) on a logarithmic scale (modified from Campocia et al., 2016).

**Figure 3 pathogens-10-00239-f003:**
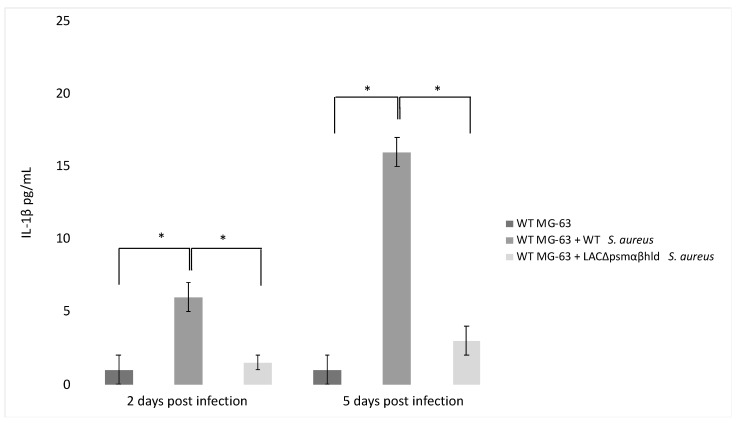
*S. aureus* phenol-soluble modulin (PSM) stimulates IL-1β release from infected osteoblast-like MG-63 cells. MG-63 cells were exposed to wild-type *S. aureus* (USA 300) and its isogenic mutant LAC Δpsmαβhld (*S. aureus* strain lacking PSMα, PSMβ and δ-toxin) at MOI 50:1. IL-1β levels were determined by ELISA 2 and 5 days postinfection. The differences were assessed by the analysis of variance (ANOVA). *p* Values  <  0.05 (*) were considered to be significant (modified from Lima Leite et al., 2020).

**Figure 4 pathogens-10-00239-f004:**
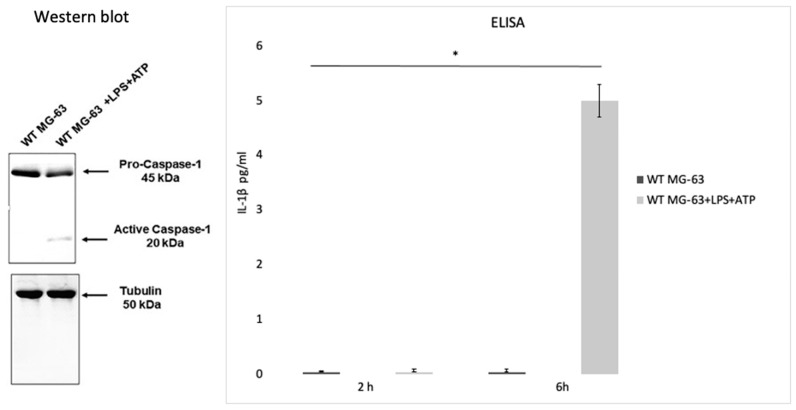
Caspase activation in the MG-63 cell line. Western blot analysis confirmed that the wild type MG-63 cells incubated with lipopolysaccharides and adenosine triphosphate (LPS+ATP) produce active caspase-1. An ELISA was performed to confirm IL-1β production in the WT MG63 cell line and in WT MG-63+LPS+ATP at 2 h and 6 h. The differences were assessed by the analysis of variance (ANOVA). *p* Values  < 0.05 (*) were considered to be significant (from and modified from Lima Leite et al., 2020).

**Figure 5 pathogens-10-00239-f005:**
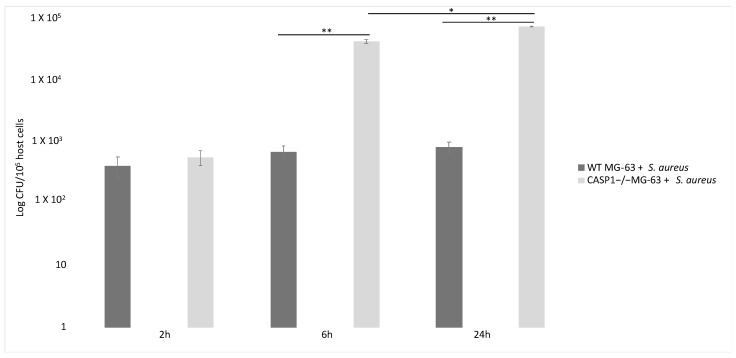
Involvement of caspase-1 in bacterial clearance in WT MG-63 and in CASP1−/− mutant MG-63 at different time points. *S. aureus* MOI 50:1. Tukey’s Honestly Significant Difference test was applied for comparison of means between the groups. *p* < 0.01 (**) for the comparison of the number of internalized bacteria in CASP1−/−MG-63 cells with those in WT MG-63 cells, and *p* values < 0.05 (*) for the comparison of the number of internalized bacteria in CASP1−/−MG-63 cells 6 and 24 h post-infection were considered to be significant (modified from Lima Leite et al., 2020).

**Figure 6 pathogens-10-00239-f006:**
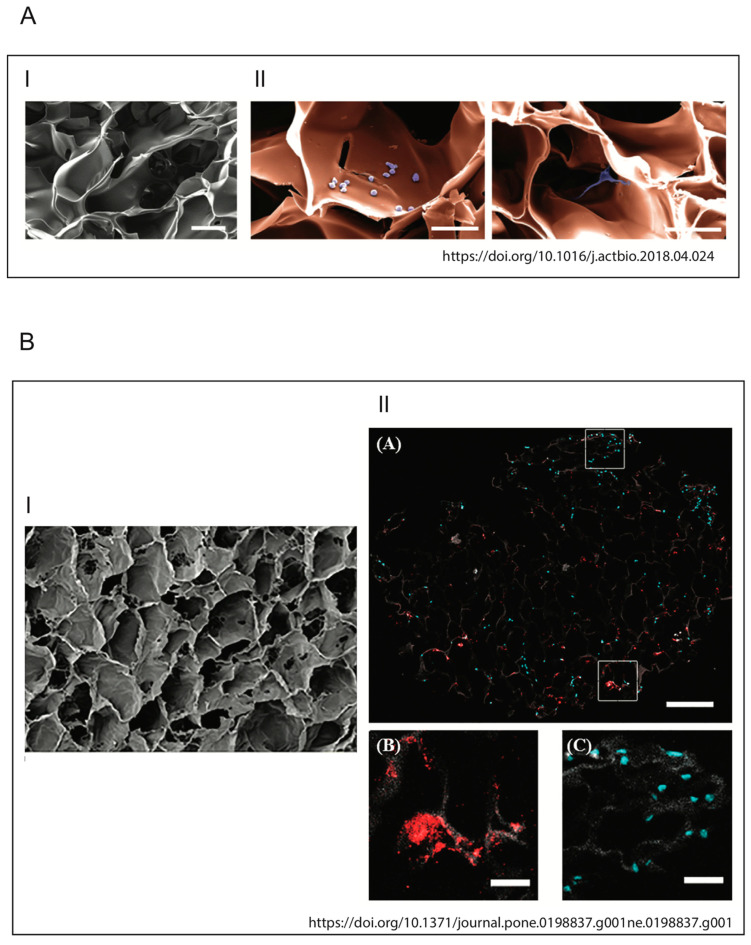
Illustrations of biomimetic 3D in vitro models. (**A**) Architecture and biocompatibility of the 3D protein scaffold. (I) Pseudocolored SEM picture of a cross section of the applied macroporous 3D protein scaffold. The image shows the porous structure inside the scaffold. Scale bar: 100 mm. (II) SEM image of 3D protein scaffold seeded with the HSPC (left) and MSC (right) cell lines. The cells (violet) are located inside of the pores of the scaffold (orange) and adhere to the scaffold material. Scale bar: 50 mm (Raic et al./Acta Biomaterialia 73 (2018) 250–262). (**B**) Pore architecture of scaffolds seeded with osteoblasts and infected by *S. aureus***.** (I) SEM of 3D scaffold only (100×, Scale bare 100 μM); (II) qualitative assessment of cellular and bacterial co-infiltration of the collagen scaffold (from Kavanagh et al., 2018).

## Data Availability

No new data were created or analyzed in this study. Data sharing is not applicable to this article.

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
