# Peer review of "Staphylococcus aureus Internalization in Osteoblast Cells: Mechanisms, Interactions and Biochemical Processes. What Did We Learn from Experimental Models?"

_pathogens, 2021, doi:10.3390/pathogens10020239_

Round 1

Reviewer 1 Report

The manuscript is of interest for the journal and it is very well structured. However there are only some points which need revision.

critics: 

I would recommend to include some highlight statements as a take home message for the reader of the journal.  

Author Response

We wish to thank you for the review. We have highlighted a take home message at the end of each paragraph inside a box, we hope they gained visibility now as you recommended.

Reviewer 2 Report

The manuscript by Stefano Stracquadanio and coworkers is a good review. It shows a strong background in the field. Some of the authors papres are autocited (but not to much). Overall the manuscript cites 137 papers. The manuscript is organized well and easy to read. Also the illustration made by A. Costantino (one of the coauthors) is brilliant.

Author Response

We wish to thanks the reviewer for his comments. 

Reviewer 3 Report

Initially, I congratulate colleagues for the beautiful and robust review. The present review manuscript provides important information about the mechanisms of intracellular survival of S. aureus. Also, it brings up-to-date knowledge about osteoblast infection by this pathogen and presents robust methodologies for the investigation of the host-pathogen mechanism. Given the importance of microbial resistance to antibiotics, the present review updates us on the mechanisms of infection of this pathogen in the host's bone tissue and brings clarity and understanding to the reader, opening new mental frontiers for further studies. That said, this rapporteur is extremely favorable to the publication of this manuscript. Suggesting only minor suggestions. Having done this, I wish the research group success and look forward to enjoying new studies. 

Suggestions:
1. In Figure 4. remove the western blot stripe to allow viewing the band immunomarked.

2. In figure 6. Only increase the images to enhance the data. 

Author Response

We wish to thanks the reviewer for his comments and suggestions. 

Now western blot in figure 4 should be more clear. 

Figure 6 is now of better quality and bigger.